# Mining Indole Alkaloid Synthesis Gene Clusters from Genomes of 53 *Claviceps* Strains Revealed Redundant Gene Copies and an Approximate Evolutionary Hourglass Model

**DOI:** 10.3390/toxins13110799

**Published:** 2021-11-13

**Authors:** Miao Liu, Wendy Findlay, Jeremy Dettman, Stephen A. Wyka, Kirk Broders, Parivash Shoukouhi, Kasia Dadej, Miroslav Kolařík, Arpeace Basnyat, Jim G. Menzies

**Affiliations:** 1Ottawa Research & Development Centre, Agriculture and Agri-Food Canada, Ottawa, ON K1A 0C6, Canada; wendy.findlay@agr.gc.ca (W.F.); jeremy.dettman@agr.gc.ca (J.D.); parivash.shoukouhi@agr.gc.ca (P.S.); kasia.dadej@agr.gc.ca (K.D.); abasn046@uottawa.ca (A.B.); 2Department of Agricultural Biology, Colorado State University, Fort Collins, CO 80523, USA; stephenwyka@gmail.com; 3USDA, Agricultural Research Service, National Center for Agricultural Utilization Research, Mycotoxin Prevention and Applied Microbiology Research Unit, 1815 N. University St., Peoria, IL 61604, USA; kirk.broders@usda.gov; 4Institute of Microbiology of the Czech Academy of Sciences CAS, 14220 Prague, Czech Republic; mkolarik@biomed.cas.cz; 5Morden Research and Development Centre, Agriculture and Agri-Food Canada, Morden, MB R6M 1Y5, Canada; jim.menzies@agr.gc.ca

**Keywords:** ergot alkaloids, ergot fungi, gene divergence, gene diversity, indole diterpenes, phylogeny, secondary metabolites

## Abstract

Ergot fungi (*Claviceps* spp.) are infamous for producing sclerotia containing a wide spectrum of ergot alkaloids (EA) toxic to humans and animals, making them nefarious villains in the agricultural and food industries, but also treasures for pharmaceuticals. In addition to three classes of EAs, several species also produce paspaline-derived indole diterpenes (IDT) that cause ataxia and staggers in livestock. Furthermore, two other types of alkaloids, i.e., loline (LOL) and peramine (PER), found in *Epichlo**ë* spp., close relatives of *Claviceps*, have shown beneficial effects on host plants without evidence of toxicity to mammals. The gene clusters associated with the production of these alkaloids are known. We examined genomes of 53 strains of 19 *Claviceps* spp. to screen for these genes, aiming to understand the evolutionary patterns of these genes across the genus through phylogenetic and DNA polymorphism analyses. Our results showed (1) varied numbers of *eas* genes in *C.* sect. *Claviceps* and sect. *Pusillae*, none in sect. *Citrinae*, six *idt/ltm* genes in sect. *Claviceps* (except four in *C. cyperi*), zero to one partial (*idtG*) in sect. *Pusillae*, and four in sect. *Citrinae*, (2) two to three copies of *dmaW*, *easE*, *easF*, *idt/ltmB*, *itd/ltmQ* in sect. *Claviceps*, (3) frequent gene gains and losses, and (4) an evolutionary hourglass pattern in the intra-specific *eas* gene diversity and divergence in *C. purpurea*.

## 1. Introduction

Fungi in the genus *Claviceps* (Clavicipitaceae, Hypocreales, Sordariomycetes) infect the florets of cereal crops, nonagricultural grasses (Poaceae), sedges (Cyperaceae), and rushes (Juncaceae) [1], followed by occupying the unfertilized ovaries and eventually replacing the seeds with fungal resting bodies, i.e., sclerotia, known as ergots [2]. In light of molecular phylogenetics, 63 named species [3,4] are classified into four sections, i.e., *Claviceps* sect. *Claviceps, C.* sect. *Citrinae*, *C.* sect. *Paspalorum*, and *C.* sect. *Pusillae*, on the basis of morphological, ecological, and alkaloid-producing features [3]. Ergot bodies or sclerotia contain a wide spectrum of alkaloids toxic to humans and animals, making them unwelcome pathogens in agricultural and food production [5,6], but also important resources for pharmaceuticals [7,8]. Among the alkaloids produced by *Claviceps*, ergot alkaloids (EAs) are the major culprit for the mass food/feed poisoning in human and livestock, as well as a number of tragedies in human history [9,10]. EAs are indole compounds characterized by a tricyclic or tetracyclic ring system. Over 80 different EAs found in nature fall into three structural groups: clavines, lysergic acid amides, and ergopeptides [8,11], corresponding to their structural complexity. Clavines are the intermediates or derivatives of the intermediates in the lysergic acid amide pathway, whereas ergopeptines are the most complex group [11]. Intensive investigations on biochemistry and molecular genetics have elucidated the EA biosynthetic pathways in EA producers especially *Claviceps* spp. [12,13]. A cluster of 12 functioning EA synthesis (*eas*) genes (*cloA*, *dmaW*, *easA, easC*–*H*, *lpsA*–*C*) in *C. purpurea* strain 20.1 were considered to encode all the enzymes needed for the end-product ergotamine and ergocryptine [14]. The four early steps, requiring *dmaW*, *easF*, *easC*, and *easE*, are responsible for the closure of the third ring resulting in chanoclavine, followed by middle steps, requiring *easD, easA, easG*, and *cloA*, for forming tetracyclic clavines, and later steps for producing the lysergic acid amides, dihydroergot alkaloids, and complex ergot peptines [13] (Figure 1). Among the 12 genes, the homologs of nine were found in *C. fusiformis* in a cluster. In *C. paspali*, two additional genes (*easP* and *easO*) were found; however, *easE* was defective. The presence or absence of *eas* genes has proven to be correlated with EA profiles in several *Claviceps* spp. and strains [13,14]. However, the investigation of *eas* gene clusters in a wide range of *Claviceps* spp. is lacking, and less is reported about the evolution of the individual gene in these clusters among and within the species.

Indole diterpenes (IDTs) are another large group of bioactive compounds with diverse structural variations, triggering toxicity in animals and insects through interfering with ion channels [16,17]. In the literature, there are copious reports that certain species in *Claviceps* (i.e., *C. paspali* and *C. cynodontis*) and close relatives in *Epichlo**ë* (*Neotyphodium* as the asexual name before implementation of the International Code of Nomenclature for algae, fungi, and plants (Shenzhen Code) [18]) produce the paspaline-derived IDTs, such as paspalitrem, lolitrem, and paxilline, causing ataxia and staggers in livestock that feed on the grasses infected by those fungal species [19,20,21]. Biosynthetic pathways and associated gene clusters of these paspaline-derived IDTs have been investigated [22,23,24], resulting in the discovery of at least 10 genes involved in IDT production in *Epichlo**ë* spp. and the prediction that *ltmG*, *M*, *C*, and *B* were responsible for the synthesis of paspaline, the basic structural backbone of IDTs, whereas *ltmP* and Q were essential for the production of lolitrem and *ltmF*, *J*, *K*, and *E*, which are required for more complex structures [25,26]. The proposed scheme for the biosynthesis of paspalitrem in *C. paspali* involved seven genes including the initial formation of paspaline through *ltmG*, *M*, *C*, and *B*, followed by the sequential functioning steps of *ltmP*, *Q*, and *F* [22]. Recently, the pre-paspaline steps were further resolved as three sequential steps: starting from *ltmG* converting farnesyl diphosphate (FPP) to Geranylgeranyl diphosphate (GGPP), followed by *ltmC* transferring GGPP to 3-geranylgeranylindole, and finally through *ltmM* and *B* yielding paspaline [27]. In addition to *C. paspali* and *C. cynodontis*, other *Claviceps* spp., i.e., *C. arundinis*, *C. humidiphila*, and *C. purpurea*, could also produce indole diterpenes or paspaline-like compounds [28,29,30]. The genome investigation of *C. purpurea* 20.1 revealed the presence of *ltmM*, *C*, *B*, *P*, *Q*, and an extra gene *ltmS* [14]. It is not known whether these genes are consistently present in various strains of *C. purpurea* and other *Claviceps* species. In addition, two other classes of alkaloids, i.e., lolines (LOL) and peramines (PER), produced by *Epichloë* spp., are known to function as insecticides, but are not associated with any toxicity symptoms in grazing mammals [31,32]. Given the close relationship between *Epichloë* and *Claviceps*, it is reasonable to raise the question of whether any of the loline or peramine gene homologs are present in any of the *Claviceps* spp. even though those two classes of alkaloids have not been reported in *Claviceps*.

The ‘hourglass model’ borrowed from ontogeny refers to the pattern that the morphological divergence of mid-development stages of an embryo are more conserved compared with earlier and later stages, resembling an hourglass with a narrow waist, but broad ends [33,34]. Before the hourglass model (HGM) was proposed in the 1990s, the early conservation model (ECM) was widely accepted, which echoed von Baer’s third law [35], i.e., embryos progressively diverge in morphology during ontogeny. The debates about these two, along with other models, i.e., adaptive penetrance model [36] and unconstrained model (random) [37], are still ongoing, although recent evidence at molecular and genomic levels has provided support for the presence of the phylotypic stage (the waist stage of development) in fungi, insects, plants, and vertebrates [38,39,40]. According to Haeckel’s biogenetic hypothesis, ontogeny recapitulates phylogeny [41]. The evident similarities between the development of an individual and the evolution of the whole biological system have been addressed by many generations [42] to verify that these models in ontogeny are recapitulated in other evolutionary processes. For example, studies on gene evolution in *Drosophila* spp. recaptured the hourglass model in that the early maternal genes showed a higher level of diversity than zygotic genes [43]. Here, we propose the biosynthesis of complex biological compounds as an analogy of the development of an organism, and ask whether any of the models fit to the evolution of the genes involved in the biosynthesis.

The objectives of the present study were to shed light on the presence of four classes of alkaloid genes (clusters) in 53 strains of 19 *Claviceps* species, and to understand the evolutionary patterns of these genes at inter- and intraspecific levels. This information helps build the foundation for future studies on chemo- and genotype associations and for developing gene-based chemotyping and toxin detection.

## 2. Results

### 2.1. Genome Assemblies

The 37 genome sequences assembled in this study resulted in 1362 to 2581 contigs, N50 values ranged from 19,946 to 55,909 bp, and the completeness measured by Benchmarking Universal Single-Copy Orthologs (BUSCO) over the fungal database (fungi odb10) ranged from 97% to 99.1% (Table 1, available in GenBank https://www.ncbi.nlm.nih.gov/ (accessed on 9 November 2021) as accessions JAIURI000000000–JAIUSS000000000 available upon publication of the article). The quality of the assemblies was equivalent to the assemblies of 17 genomes from previous studies (Table 1) [44,45]. Overall, the 54 assemblies of 53 strains (two versions of assemblies for CCC1102 were included because certain genes were obtained from one or the other assemblies) of 19 *Claviceps* species included in this study belong to three sections: *C.* sect. *Citrinae*, *C.* sect. *Claviceps*, and *C.* sect. *Pusillae*, from six continents (Africa, Asia, Australia, Europe, North America, and South America) and on host plants in 26 genera (Appendix A).

### 2.2. Presence of Four Classes of Alkaloid Genes in 53 Genomes

One thousand sequences of 19 loci were extracted from the 53 genome assemblies as detailed below. The DNA sequences of each genes were submitted to Genbank associated with accession numbers: *cloA* (49 sequences) MZ882098–MZ882146, *dmaW* (118 sequences) MZ871640–MZ871757, *easA* (51 sequences) MZ851397–MZ851447, *easC* (50 sequences) MZ851807–MZ851856, *easD* (51 sequences) MZ871767–MZ871817, *easE* (66 sequences) MZ877968–MZ878033, *easF* (88 sequences) MZ881959–MZ882046, *easG* (50 sequences) MZ882047–MZ882096, *easH1* (50 sequences) MZ934760–MZ934809, *easH2* (32 sequences) MZ934810–MZ934841, *lpsB* (48 sequences) MZ934842–MZ934889, *lpsC* (44 sequences) MZ934890–MZ934933, *idt*/*ltmB* (55 sequences) MZ935033–MZ935087, *idt*/*ltmC* (47 sequences) MZ935088–MZ935134, *idt*/*ltmG* (three sequences) MZ935227–MZ935229, *idt*/*ltmM* (47 sequences) MZ935135–MZ935181, *idt*/*ltmP* (46 sequences) MZ934987–MZ935032, *idt*/*ltmQ* (60 sequences) MZ934934–MZ934986, and *idt*/*ltmS* (45 sequences) MZ935182–MZ935226.

#### 2.2.1. Ergot Alkaloid Genes (eas)

More consistency in terms of presence/absence of *eas* genes was observed in *C.* *sect. Claviceps* than *C*. sect. *Pusillae.* The results from BLASTn searches using in-house script (see Section 4.2 for details) and Geneious mapping (https://www.geneious.com, accessed on 9 November 2021) with reference genes showed the genomes of all isolates from *C.* sect. *Claviceps* contained at least 10 *eas* genes matching the *C. purpurea* 20.1 reference sequences (Table 2; *lpsA1* and *lpsA2* were excluded from analyses as they were heavily fragmented due to the significantly long length. A study on long-read sequencing of several selected strains by Hicks et al. was focused on these two genes (in this Special Issue). The 10–12 genes were assembled on two to three contigs. For most strains, nine genes (*lpsC, easA, lpsB, cloA,* and *easC*-*G*) were on the same contig. Genes after *dmaW*, i.e., *easH1*, *easH2*, and fragments of *lpsA*, were on different contigs. The *easH2* gene was either not detected or on a separate contig possibly due to the long length of *lpsA1* because it was located between *lpsA1* and *A2* in the reference genome *C. purpurea* 20.1. The exceptions were *C. humidiphila* LM576*, C. spartinae* CCC535, *C. purpurea* LM461, and *C. ripicola* LM220 and LM454, in which *lpsC* was on a different contig, or *lpsC* along with the next three to four genes were on the same contig separated from other genes (Table 2).

Both inter- and intraspecific variation was observed, regardless of the general consistency of presence of *eas* genes. Species-specific features included all three strains of *C. occidentalis* have two partial copies of *dmaW* (~658 bp, ~641 bp composed of a partial exon 1 and full-length exon 2 and 3) and a single copy of all other *eas* genes except *easH*. Of a relevant note, all partial genes detected in the present study were located at the end of contigs. Moreover, all three strains of *C. quebecensis* had a second partial nonfunctioning copy of *easE* (275, 275, and 1208 bp) and two partial copies of *easF* with good open reading frames (ORFs), and they were lacking *easH2* (Table 2).

Intraspecific variation among the 27 strains of *C. purpurea* was evident as most strains contained one copy of *lpsC*, *easA*, *lpsB*, *cloA*, *easC*, *D*, *G*, *H1*, and *H2*, and two copies of *dmaW*. However, three strains (LM65, LM72, and LM582) lacked *easH2*. Eleven strains had a second copy of *easE* (*easE2*), six full- or near-full-length and five partial, but these gene fragments contained indels of various sizes and internal stop codons (Table 2). This would indicate that they may not be functional genes unless those variations were caused by sequencing or assembly errors. In contrast, the second copy of full-length *easF* (*easF2*) from LM72 (MZ881984) and LM461 (MZ881981) had good ORFs. The *easF2* gene of the other six strains was split on two contigs with gaps in the middle. Most of these fragments, except the second exon at the 3′ end of Clav26, Clav55, and LM470, were free of internal stop codons. Four strains had a full length (or close to full length), and one strain (LM469 652 bp) had a partial third copy, *easF3*, yet these gene fragments had a number of indels and internal stop codons (Table 2). The intraspecific variations were also found in *C. arundinis* and *C. ripicola* (Table 2).

The six genomes from *C.* sect. *Pusillae* had more variable numbers of the *eas* genes observed, but all six genomes lacked *lpsC* and *easH2* (Table 2). The strain *C. lovelessii* CCC647 had the highest number of matches, i.e., 10 full- or near-full-length matches (*cloA* 1788 bp, *easD* had an 8 bp short gap at split region), while all but *easH1* and *lpsB* had good ORFs. In contrast, *C. digitariae* CCC659 had only two gene matches: *dmaW* and *easA*, but both were full-length with good ORFs. *C. maximensis* CCC398 and *C. citrina* CCC265 (*C.* sect. *Citrinae*) had no matches for any *eas* genes (Table 2).

Examining each *eas* genes, *easA* was present the most consistently in 51 of 53 genomes as a single copy and had good ORFs, except for the one in *C. pusilla* CCC602 which had an internal stop codon. Similarly, *lpsB*, *cloA*, *easC, easD*, and *easG* were present as a single copy in all species of sect. *Claviceps* and two to four species in sect. *Pusillae* (Table 2).

For *easE*, all species in sect. *Claviceps* contained at least one copy, six strains of *C. purpurea* (LM39, LM63, LM72, LM461, LM469, and LM474)) had a full length second copy (*easE2*), and the other five strains *C. purpurea* (Clav04, Clav46, Clav52, Clav55, and LM470), all three *C. quebecensis*, one *C. spartina*, and one *C. monticola* had a second partial copy. Compared with the *C. purpurea* 20.1 *easE1* reference sequence, all the *easE2* sequences contained a large number of deletions (gaps) of various sizes in exon and intron regions, internal stop codons, and no start codon, indicating that they are likely not functional. For species in sect. *Pusillae*, one copy of *easE* was found in four species with good ORFs (*C. africana* CCC489, *C. lovelessii* CCC647, *C. pusilla* CCC602, and *C. sorghi* CCC632).

For *easF*, all species in sect. *Claviceps* contained at least one copy; however, two strains of *C. purpurea* (Clav55 and LM470) had internal stop codons near the 3′ end. Twenty-three strains of seven species (*C*. *arundinis*, *C. humidiphila, C. monticola, C. pazoutovae, C. purpurea, C. quebecensis*, and *C. spartinae*) had a second full-length or partial copy, among which 19 strains had good ORFs. In addition, a third copy was found in some *C. purpurea* strains in full length (LM39, LM63, and LM65) or partial (LM461 and LM469). Even though with 77–93% similarity to *C. purpurea* 20.1 *easF1*, none of the third copies had a correct open reading frame (not functional) (Table 2). Three species in sect. *Pusillae* (*C. africana*, *C. lovelessii*, and *C. sorgji*) had one functioning copy.

For *dmaW*, most species (strains) in sect. *Claviceps* contained two full-length copies or copies split on two contigs with gaps. Six strains of *C. purpurea* (Clav26, Clav52, LM223, LM232, LM4, and LM470) had a partial second copy, but all three strains of *C. occidentalis* had partial sequences (~650 bp) for both copies. One strain of *C. arundinis* (CCC1102) had a third copy in full length, with 81% and 83% similarities with *dmaW1* and *dmaW2*, but frameshifts and internal stop codons were present. Five species in sect. *Pusillae*, except *C. maximensis,* had one copy.

Interestingly, the additional copies of *easE, easF*, and *dmaW* were more or less clustered together, such that the second copies of all three genes were present on the same contig in *C. monticola* CCC1483 and *C. spartinae* CCC535 (Figure 2A). Alternatively, the *easF2* sequence was split on two contigs, which were located with *easE2* on one contig and *dmaW2* on the other, i.e., *C. purpurea* Clav55, and *C. quebecensis* Clav32, Clav50, and LM458 (Figure 2B). More commonly, *easE2* was on the same contig as *easF2*, whereas *dmaW* was on another contig, such as in seven strains of *C. purpurea* (Clav46, Clav52, LM470, LM474, and LM72; Table 2), or *easF2* co-located with *dmaW* when *easE* was a single copy (LM583; Figure 2C). In cases when the third copy of *easF* was present, they were often on the same contig with *dmaW2,* i.e., *C. purpurea* LM39, LM63, LM65, and LM469 (Figure 2D). The arrangement in LM461 was more peculiar in that the second copies of *easE* and *easF* were on the same contig with dmaW1 and *easG* (a single-copy gene), which indicates that they may all be on the primary ergot alkaloid gene cluster (Figure 2E). The third *dmaW* from CCC1102 (from SW assembly) was not connected to other *eas* genes (Table 2).

For *easH*, *easH1* was present in 50 genomes, except *C. citrina*, *C. digitariae*, and *C.maximensis*; however, the genes of the four species (CCC489, CCC602, CCC632, and CCC647) in *C*. sect. *Pusillae* had numerous indels of various sizes throughout the sequence, causing frameshifts and internal stop codons. Further validation of the sequences is needed to confirm whether these are functioning. The *easH2* gene was present in 32 strains of six species (*C. arundinis*, *C. humidiphila*, *C. occidentalis*, *C. perihumidiphila*, *C. purpurea*, and *C. ripicola*). The reference sequence of *easH2* from *C. purpurea* 20.1 was 840 bp, which is about 100 bp shorter than *easH1* (945 bp), and it was considered a pseudogene. Our results showed that the 32 *easH2* sequences had variable lengths and high levels of nucleotide variation (see more notes in later sections: phylogenies and gene diversity). Most of these sequences appeared not functional; however, the lengths of the sequences from two strains of *C. ripicola* (LM218 and LM220) were 954 bp and contained full-length ORFs, indicating that they are likely functioning genes.

For *lpsC*, at least one strain per species in sect. *Claviceps* (except *C. perihumidiphila*) showed one copy of *lpsC*, i.e., in total, 43 out of 46 strains contained a single copy of *lpsC*, among which three strains of *C. purpurea*, i.e., Clav26, LM4, and LM232, had a single internal stop codon; otherwise, the full range of sequences aligned very well with the reference. It is possible that the single internal stop codon could be a sequencing error. Another five strains/species, including *C. capensis* CCC1504*, C. cyperi* CCC1219*, C. humidiphila* LM576, *C. monticola* CCC1483, *C. purpurea* LM223, and *C. spartinae* CCC535 had partial sequences 1000–5000 bp long. These sequence fragments contained several indels and internal stop codons, and they are apparently not functional genes. Only one strain of *C. perihumidiphila* lacked *lpsC*.

#### 2.2.2. Indole-Diterpene/Lolitrem (idt/ltm) Genes

Compared with *eas* genes, the presence/absence and copy numbers of *idt/ltm* genes were less variable. Through mapping genome assemblies to the reference genes, all members in sect. *Claviceps* had one copy of *ltmC*, *M*, *P*, and *S* and one or two copies of *idt/ltmB* and *Q*, except *C. cyperi* CCC1219 that lacked *ltmQ* and *S*. All members in *C.* sect. *Pusillae* had no matches to any *ltm* genes, whereas members of sect. *Citrinae* (*C. citrina* CCC265) had full-length matches with *ltmB*, *C*, *G*, and *M*.

Notable species-specific features were that all three strains of *C. occidentalis* (LM77, LM78, and LM84) had two partial copies *ltmQ* (1517–1518 bp); *C. arundinis* (CCC1102, LM583), *C. perihumidiphila* (LM81), *C. ripicola* (LM218, LM219, LM220, and LM454), and *C. spartinae* (CCC535) had two functional copies of *ltmB* (Table 3). The translated sequences of *ltmS* from three strains of *C. occidentalis* (LM77, LM78, and LM84) and three strains of *C. quebecensis* (Clav32, Clav50, and LM458) were 14 amino-acid residues longer than other species, and those 14 amino acids were identical among the six strains.

Intraspecific variations were observed in *C. purpurea*; four out of 27 strains showed a second copy of *ltmQ* (Table 3). In the strain Clav04, the fragment on the primary cluster (contig130) *ltmQ1* was a partial copy, whereas another copy on contig 637 was a full-length copy (*ltmQ2*) with a good ORF. Clav46 had two partial copies; ironically, the copy on contig 43 (where all other *ltm* genes co-located) had a number of short deletions causing frameshifts and internal stop codons, whereas the copy on contig 229 had good ORFs, except that the first 243 bp (including 53 residuals in exon 1 and partial exon 2) were missing. On the other hand, some of the single-copy *ltmQ* sequences, such as in *C arundinis* CCC1102, *C. pazoutovae* CCC1485, *C. perhumidiphila* LM81, C*. purpurea* LM72, *C. quebecensis* Clav32 and LM458, *C. ripicoloa* LM218 and LM219, and *C. spartinae* CCC535, had varied number of indels causing frameshifts and internal stop codons; however, phylogenetically, they still belonged to copy 1 (more details in in Section 2.3).

All six genes were clustered on the same contig in 29 strains of the 12 species in sect. *Claviceps*; otherwise, at least three genes were on the same contig. The clustered six *ltm* genes were arranged in the same order as in *C. purpurea* 20.1 [14] (Table 3; gene coordinates are not shown). In *C. citrina*, *ltmB* and *C* were on the same contig (1947), whereas *ltmM* and *G* were on separate contigs. It is not assessable whether they were in one cluster. In general, the inter-gene sequences ranged from 500–1200 bp; however, several strains had very long spaces between *ltmP* and *B*, such as 4 kb in *C. ripicola* LM220 and over 2 kb in LM218 and *C. arundinis* CCC1102 and LM583 (results not shown).

Through the additional BLAST searches with lower stringency (E-value < E^−50^), fragments of 483 and 501 bp of *ltmG* from *C. maximensis* CCC398 and *C. digitariae* CCC659, respectively, were pulled out by using *ltmG* from *C. paspali* RRC-1481. They were 76% and 78% similar, respectively, to the reference sequence in the coverage (comparable to the 74% similarity between *C. citrina* CCC265 and *C. paspali* RRC-1481). Running BLAST searches of these two fragments to the NCBI database indicated that 60 bp of the 483 bp from *C. maximensis* matched with *Beauveria bassiana* ARSEF 2860 geranylgeranyl pyrophosphate synthetase; 279 of 501 bp from *C. digitariae* matched with *idtG* (geranylgeranyl diphosphate synthase) from *Periglandula ipomoeae* strain IasaF13.

#### 2.2.3. Loline Alkaloid (lol) and Peramine (per) Genes

All the searches with *lol* and *per* reference genes resulted in no hits, except for the low-stringency BLAST with *lolC* that resulted in small fractions of sequences (~150–180 bp) matched with the start of the fifth exon for seven species (strains): *C. africana* (CCC485), *C. citrina* (CCC265), *C. digitariae* (CCC659), *C. lovelessii* (CCC647), *C. maximensis* (CCC398), *C. pusilla* (CCC602), and *C. sorghi* (CCC632). These fragments matched with 80% to 92% identity to *O*-acetylhomoserine from *Purpureocillium lilacinum* (XM 018324292)*, Drechmeria coniospora* (XM 040800194), and *Verticillium dahliae* (XM 009654023) in the NCBI database https://blast.ncbi.nlm.nih.gov/Blast.cgi accessed in August 2021. These sequences were not submitted to GenBank because of their short length.

### 2.3. Phylogenies of eas and idt/ltm Genes

The individual phylogenetic trees of 11 *eas* genes all agreed on the long-branched separation between *C.* sect. *Pusillae* and sect. *Claviceps,* which was congruent with the pattern inferred by the previous multigene analyses combined with morphological, ecological, and metabolic features [3] and supported by the phylogenomic analyses [44] (Figure 3a). In *C.* sect. *Pusillae*, all genes agreed on the close proximity of *C. fusiformis*, *C. lovelessii,* and *C. pusilla*, as well as of *C. africana* and *C. sorghi.* The main incongruence among the gene trees appeared in the uncertain placements of *C. digitariae* and *C. paspali*, as well as the variant relationships among *C. fusiformis*, *C. lovelessii*, and *C. pusillae,* which could be a result of insufficient sampling (see further explanation in Section 3; Figure 3b–d and S1).

In terms of the species relationships in the sect. *Claviceps*, considering single-copy genes, a majority of gene trees agreed on the grouping of the four major clades inferred by the previous phylogenomic study [44]. For communication convenience, we named them as four Batches to avoid confusion with species level and general use of clades: Batch humidiphila including *C. arundinis*, *C. humidiphila*, and *C. perihumidiphila*, Batch purpurea including *C. capensis*, *C. monticola*, *C. pazoutovae*, and *C. purpurea* (previously designated as Clade purpurea by Píchová et al. [3]), Batch occidentalis including *C. occidentalis* and *C. quebecensis*, and Batch spartinae including *C. ripicola* and *C. spartinae* (Figure 3a and Appendix A). The exceptions were *C. perihumidiphila* and *C. cyperi* that had uncertain placement on different gene trees (Appendix Ab,d,f,g). The more notable disparities among the gene trees appeared in the order of divergence of the four Batches from C. sect. *Pusillae* or sect. *Paspalorum* (Figure 4 and Appendix A and Appendix A). Previous phylogenomic analyses resulted in the topology of a twice bifurcate pattern, ((Batch humidiphila)(Batch spartinae); (Batch occidentalis)(Batch purpurea)) [44], and this pattern was only supported by *easG* (Figure 4a). A slight variation of the *easA* tree appeared in that Batch humidiphila was an earlier diverged lineage than Batch spartinae, and these two formed a paraphyletic group instead of a monophyletic group (Figure 4b). All other genes supported the derived position of Batch humidiphila and Batch spartinae (Figure 4c–e). Furthermore, eight genes (*cloA*, *dmaW1*, *easC*, *easE1*, *easH1*, *lpsC*, and *ltmB1*) placed Batch purpurea at a more ancestral position than Batch occidentalis, whereas six genes (*easF1*, *lpsB*, *ltmM*, *ltmP*, *ltmS*, and *ltmQ1*) reversed the divergence order of these two Batches (Figure 4c,d). The other three genes (*easD*, *lpsC*, and *ltmC*) showed an unresolved order of divergence (Figure 4e).

As for genes with multiple copies, the most complex was *dmaW*. The *dmaW2* sequences were separated into two groups. Group I included 16 strains of eight species (all non-*C. purpurea dmaW2* except *C. monticola* CCC1483 and *C. pazoutovae* CCC1485), forming a parallel lineage with their *dmaW1* counterpart and representing one gene duplication at node ① (Figure 5a and Appendix Aa). Group II included *C. purpurea*, *C. monticola* CCC1483, and *C. pazoutovae* CCC1485, as well as one strain of each *dmaW1* (LM60) and *dmaW3* (CCC1102). This group diverged from *C. purpurea dmaW1*, representing the second duplication at node ②. Within group II, the otherwise consistent close relationship between *C. monticola* and *C. pazoutovae* was broken by seven strains of *C. purpurea*. This can be explained by a third duplication at node ③. The presence of *dmaW3* of *C. arundinis* CCC1102 and *dmaW1 C. purpurea* LM60 in group II indicated extra duplication events at nodes ④ and ⑤ (Figure 5a).

The second and third copies of *easF* (*easF2*, *easF3)* grouped in one clade diverged from *C. cyperi easF1*. Within this clade, *C. purpurea easF2* (14 strains) appeared as a paraphyletic group, from which diverged a clade composed of *C. purpurea easF3* (five strains) and a subclade *easF2* of *C. quebecensis*, *C. humidiphila*, *C. arundinis*, *C. spartinae*, *C. pazoutovae*, and *C. moticola.* From this tree topology, at least two gene duplication events were inferred (Figure 5b and Appendix Ab).

The second copy of *easE* (*easE2*) from 16 samples grouped into one clade, which diverged from *easE1* of *C. occidentalis*. However, within the *easE2 c*lade, *C. purpurea* samples were separated into two subclades. The sample Clav 04 appeared as an orphan clade located close to *C. quebecensis easE2*, and another 10 samples grouped together and had affinity with *C. monticola easE2*, indicating that the historical gene duplications possibly occurred twice at nodes ① and ② (Figure 5c and Appendix Ac).

The second copies of *easH* (*easH2*) were grouped into three groups that diverged three times independently. Group I includes two strains of *C. ripicola* (LM218 and LM220) that diverged from *easH1* of the clade composed of *C. capensis*, *C. moticola*, and *C. pazoutovae*. As noted earlier, the sequence lengths of *easH2* from these two strains are similar to *easH1* and contained good ORFs, indicating that they were likely from a very recent gene duplication. Group II, including three strains of *C. occidentalis*, one strain each of *C. arundinis*, *C. humidiphila*, and *C. perihumidiphila*, and 15 strains of *C. purpurea*, diverged from the *easH1* clade composed of eight species in sect. *Claviceps* (*C. occidentalis*, *C. cyperi*, *C. quebecensis*, *C. perihumidiphila*, *C. ripicola*, *C. spartinae*, *C. arundinis*, *C. humidiphila*, and *C. purpurea*). Group III, including nine strains of *C. purpurea* and the reference sequence of *C. purpurea 20.1 easH2*, diverged within the clade of *C. purpurea easH1* (Figure 5d and Appendix Ad).

For *idt/ltm* genes, the second copies of *ltmB* can be considered as one group arising from one gene duplication, except that *ltmB1* of *C. humidiphila* LM576 was placed in this group. This sequence was the only copy detected in LM576 and, therefore, labeled as copy one. However, it was on a separate contig (contig 478), clustered with neither *ltmP* and *ltmQ* (contig 945, Table 3) nor *ltmC, ltmS*, and *ltmM* (contig 745). It is very likely that this represents the second copy of this gene, and copy one was either lost or not detected (Figure 5e and Appendix Ae).

The three partial *ltmQ2* genes from three strains of *C. occidentalis* grouped closely with a clade composed of four strains *C. purpurea ltmQ1* (Clav04, Clav46, LM71, and LM72) and two *ltmQ2* (Clav55, and LM461) (Figure 5f and Appendix Af). As noted earlier in Section 2.2.2, *ltmQ1* of Clav04 and Clav46 was either a partial gene or a nonfunctional gene, respectively, whereas the second copies were functioning genes. Here, *ltmQ2* of Clav04 and Clav46 grouped in *C. purpurea ltmQ1* clade 1. This situation can be explained by a scenario in which these two copies might have switched locations due to errors in assembling. For another two sequences, *ltmQ1* of LM71 was on a different contig with other *ltm* genes, and in LM72, the gene was split into two contigs, where one half was connected with *ltmP*, while the other half was independent. Overall, these four sequences appeared as the same copy in *C. purpurea ltmQ2* (Clav55 and LM461). If that is the case, one gene duplication event possibly happened at node ①. Alternatively, the *ltmQ2* of Clav04 and Clav26, as well as the two *ltmQ2* groups, could have resulted from independent gene duplications (Figure 5f and Appendix Af). Long-read sequencing, i.e., Nanopore or PacBio, could bring more insight by ruling out the possible assembly errors.

### 2.4. Intraspecific Genetic Variation within C. purpurea

Overall, the haplotype diversities (Hd) of *eas* genes ranged from 0.936 to 1 (close to saturation), except for *easH2* that had a lower value, 0.858. Nucleotide diversity (Pi) of *eas* genes ranged from 0.08 (*easD*) to 0.168 (*easH2*), the average number of nucleotide difference (K) ranged from 7.1510 (*easD*) to 212.238 (*easE2*), tree-based divergence from COT ranged from 0.06 (*easA* and *easD*) to 0.150 (*easH2*), and tree-based diversity ranged from 0.01 (*easD*) to 0.219 (*easE2*). In general, *easD* and *easA* had lower values for divergence and/or diversity. The second copies of *dmaW*, *easE*, *easF*, and *easH* had much higher values of the four parameters. Some of those genes may not function and, therefore, had fewer functional constraints. If only the first copy of the genes was considered, the genes with the highest diversity and divergence values were Pi 0.03 (*dmaW1*), K 92.379 (*lpsC*), tree-based divergence from COT 0.0025 (*dmaW1*), and tree-based diversity 0.038 (*dmaW1*). The two genes functioning in the middle of the pathway, i.e., *easA* and *easD*, were observed to be the most conserved genes compared with the other genes in the earlier or later steps (Table 4, Figure 6a).

^1^ Sequences with large gaps causing a significant reduction in the number of sites were excluded from the analyses. ^2^ Tree-based divergence from the center of tree (COT) and diversity were estimated by DIVIEN; other parameters were estimated by DnaSP.

Compared with the first copy of *eas* genes, *idt*/*ltm* genes had a similar level of the highest diversity and divergence. Pi ranged from 0.007 (*ltmM* and *ltmS*) to 0.02 (*ltmQ1*), average number of nucleotide difference (K) ranged from 6.839 (*ltmS*) to 41.486 (*ltmQ1*), tree-based divergence from COT ranged from 0.005 (*ltmM*) to 0.066 (*ltmB1*), and tree-based diversity ranged from 0.009 (*ltmM*) to 0.04 (*ltmQ*) (Table 4, Figure 6b).

## 3. Discussion

### 3.1. Correlations between the Presence/Absence of Alkaloid Genes and Alkaloid Production

It has been shown while attempting to induce EA production for pharmaceutical purposes (see review by Flieger [46]) that different ergot species produce varied types of ergot alkaloids. Simultaneously, mycologists explored the use of alkaloid chemistry for characterizing *Claviceps* species [47,48]. Pažoutová and colleagues [49] differentiated chemoraces using the qualitative and quantitative features of EA production. A systematic study on EA production in 43 *Claviceps* species confirmed that ergopeptides were produced only by the members in *C.* sect. *Claviceps*, whereas dihydroergot alkaloids (DH-ergot alkaloids) were produced only by certain members of *C.* sect. *Pusillae*, i.e., *C. africana*, *C. gigantea*, and *C. eriochloe.* Sixteen out of 28 species in *C.* sect. *Pusillae* were shown not to produce any EAs, including *C. maximesis*, *C. pusillae*, and *C. sorghi*. Species only producing clavines included *C. fusiformis*, *C. lovelessii*, and three other species [3]. More recent studies demonstrated that the indole alkaloid profiles supported the recognition of new species based on molecular and ecological data [29,30].

The EA genes detected in the present study were consistent with the known EA production of the included species, for the most part. For example, *C. africana* CCC489 had eight genes detected (lacking *cloA*, *easH2*, *lpsB*, and *lpsC*), and all appeared to be functional, consistent with its production of DH-ergot alkaloids. Similarly, in *C. lovelessii* CCC647, ten EA genes were detected (lacking *lpsC* and *easH2*); however, *easH1* and *lpsB* had mutations resulting in a number of internal stop codons, which is consistent with the production of *clavines*, a product of the early pathway [3]. A lack of EA production corresponded to no matches for any EA genes in *C. maximensis* CCC398 and *C. citrina* CCC265 (*C.* sect. *Citrinae*). However, for *C. pusillae* and *C. sorghi*, several functional genes were detected even though no EA production was reported [3]. In *C. pusillae* CCC602, eight genes had full-length matches (*dmaW1*, *easA*, *C*, *D*, *E*, *G*, and *H1*, and *lpsB*) and one partial match (*cloA* 332 bp)*,* but only *dmaW1*, *easC*, and *easE* had ORFs. The lack of *easF*, the second step in the pathway encoding dimethylallyltryptophan *N*-methyltransferase, might explain the lack of production of EAs. *C. sorghi* CCC632 had seven full-length matches (*dmaW1* and *easA*, *C*, *E*, *F*, *G*, and *H1*) and two partial (*cloA* 435 bp and *easD* 653 bp). Except for *cloA* and *easH1,* all other genes had good ORFs. Theoretically, at least chanoclavine should be produced unless those genes were not expressed possibly due to a lack of triggers from physical or environmental conditions [50].

Only the members in *C.* sect *Claviceps* had *lpsC* and *easH2*, although *C. perihumidiphila* LM81, one strain of *C. ripicola* (LM454), and *C. arundinis* (LM583) lacked *lpsC*, and *C. capensis*, *C. cyperi*, *C. humidiphila*, and *C. monticola* had a partial *lpsC*. Moreover, three *C. purpurea* strains (LM65, LM72, and LM582) and three *C. quebecensis* strains (Clav32, Clav50, and LM458) lacked *easH2.* Whether the absence of these genes causes variations in their EA profiles requires a systematic investigation on the associations between *eas* genes and products in those species. It is worth noting, however, that the possibility of false negatives in genome screening cannot be ruled out. For instance, for *C. arundinis* CCC1102, *lpsC* was detected in the WF version of the genome assembly (created in the present study), but not in the previous version (SW [44], Table 2). The opposite also occurred in that a full length of *dmW3* was detected in SW assemblies, but only partially (360 bp) in WF assemblies (this study).

The production of indole diterpenoid compounds in ergot fungi was reported in a small number of species, i.e., *C. arundinis*, *C. cynodontis*, *C. humidiphila*, *C. paspali*, and *C. purpurea* [21,28,29,30]. Our genome mining showed that *ltmQ*, *P*, *B*, *C*, *M*, and *S* were present in all species in *C.* sect. *Claviceps* except *C. cyperi*. Furthermore, *ltmB*, *C*, and *M* and a nonfunctioning *ltmG* were detected in *C. citrina,* while a partial *ltmG* was detected in *C. maximensis* CCC389 and *C. digitariae* CCC659. According to the proposed pathway, to produce paspaline, the first step requires *ltmG*, followed by *ltmC*, *ltmM*, and *ltm**B* [27]. The absence of *ltmG* could stop production unless GGPP is present through other resources. This might be the case in the producers of indole diterpenoid compounds listed above. In the same way, it is very likely that most of the species in C. sect. *Claviceps* and the three species in sect. *Citrinae* and sect. *Pusillae* could also produce some forms of indole diterpenoid compounds.

### 3.2. Macro-Evolution of the Gene Clusters—Frequent Gene Duplications and Losses

Ergot alkaloid diversity among diverse producers, i.e., species in Hypocreales, Eurotiales, and Xylariales, was formed by three major processes: gene gains, gene losses, and gene sequence changes [13,14]. This is true within the genus *Claviceps.* A recent genus-level genome comparison hypothesized that unconstrained tandem gene duplications were caused by putative loss of repeat-induced point mutations in *C.* sect. *Claviceps* [44]. This pattern of duplication was confirmed here by the presence of a cluster of second or third copies of *easE*, *easF*, and *dmaW*, as well as second copies of *ltmQ* and *B* (Table 2 and Table 3). Moreover, *easE2* and *F2* of *C. purpurea* LM461 were on the same contig as *easG* and partial *dmaW1,* suggesting that the second copies of *easE* and *F* were arranged on the primary cluster possibly as a result of tandem gene duplication. None of the extra gene copies were found in *C*. sect. *Pusillae* or sect. *Citrinae*, consistent with a previous observation that the genomes of sect*. Pusillae* and sect. *Citrinae* had much fewer gene duplication events predicted [44]. According to the phylogenies of multicopy genes, one to five gene duplications can be inferred for individual genes. The *dmaW* gene, encoding the enzyme for the first and determinant step of *EA* production, had the highest number of potential gene duplications. Even though the presence of *dmaW* was conserved across various EA producers and proven to be a monophyletic group [51], its evolutionary rate was faster than genes in the middle steps of the EA pathway.

Gene losses can be inferred through the discrepant placement of certain gene copies on the phylogenies. For instance, one copy of *ltmB* in *C. humidiphila* LM576 was detected; however, this copy grouped with *ltmB2*. It is very likely that this was the second copy of *ltmB* gene, and the first copy was either lost or not detected (Figure 5e, see also Section 2.2.2. and Section 2.2.3). The *ltmQ1* from four strains of *C. purpurea* (LM71, LM72, Clav04, and Clav46) was placed in the *ltmQ2* clade. For LM71 and LM72, there was only one copy detected (*ltmQ1*); the scenario is likely similar to *ltmB* of LM576, where this single copy was the second copy, and the original gene was either lost or not detected (Figure 5f). On a related note, *ltmQ2* of Clav04 and Clav46 was located in the *ltmQ1* clade. An intuitive explanation would be that the identities of the two copies switched due to assembly artefacts (Figure 5f). Lastly, the incongruent order of divergence of the four Batches of species in *C.* sect*. Claviceps* inferred by single-copy genes could be explained as lineages sorted during the frequent gains and losses of the ancestral genotypes (Figure 4). Unlike *C.* sect. *Claviceps*, the phylogeny incongruence in *C.* sect *Pusillae* was mainly caused by the uncertain placement of *C. digitariae* and *C. paspali.* In light of the genome structure, this was likely caused by insufficient sampling instead of gene lineage sorting.

### 3.3. Micro-Evolution of eas Genes within C. purpurea—An Approximate Hourglass Model

The inter- and intraspecific variations of the second metabolite gene clusters in fungi are typically reported as variations in structures, gene contents, copy numbers, null alleles, and nonhomologous clusters (see review by Rokas [52]). Fewer studies have focused on the DNA sequence variations in each of the gene members. Lorenz et al. [53] identified the sequence differences in *lpsA* between two *C. purpurea* strains (P1 and ECC93) that were associated with the different alkaloid types; however, they could not find differences in *cloA* between *C. fusiformiis* and *C. hirtella* that could explain why this gene was functional in the former but not in the latter. Phylogenetic analyses of DNA sequences of four core genes (*dmaW*, *easF*, *easC*, and *easE*) from selected samples across Clavicipitaceae (with emphasis on *Epichlo**ë*) uncovered extensive gene losses, and the origin of EA clusters on Clavicipitaceous fungi was determined to be direct descent rather than horizontal transfer [13].

The present study is the first, to our knowledge, to examine the variations of each gene on a fine scale, i.e., among 28 strains of *C. purpurea*. Both DNA polymorphism analyses of the DNA sequence alignments through DnaSP and tree-based diversity and divergence analyses using the DEVIEN software indicated that the evolutionary rate of early step genes, i.e., *dmaW* and *easF* is much higher than the middle step genes, i.e., *easA*, *C*, *D*, and *E* (Figure 6, Table 4). The pattern matches with the hourglass model in ontogeny, which was also evidenced in genomic studies [39]. The hourglass model (HGM) and early conservation model (ECM) in ontogeny are explained by developmental constraints. HGM considers that, at the middle stage, the *meta*- and *cis*-interactions reach the highest complexity, posing constraints for development [54,55], whereas ECM considers the constraints at early stage to be critical because any alterations at early stage would cause cascading effects [56]. The EA pathway was reported as an unusually inefficient one such that a high volume of certain intermediates were accumulated more than needed for producing the end-products [57]. This may impose less selective pressure on the middle steps. The sclerotia of *C. purpurea* from tall fescue contained chanoclavine (4 ± 3 µg/g) and agroclavine (2 ± 1 µg/g) in addition to the end-products, i.e., ergopeptines and ergnovine [57]. The extra amount of chanoclavine coincides with the lowest evolutionary rates of *easD* and *easA* inferred in the present study (Figure 6). The role of *easD* is to oxidize chanoclavine to chanoclavine aldehyde, followed by the reactions of *easA* and *easG* to yield agroclavine. It is likely that *easD* is under less selective pressure because plenty of supplies are available. Alternatively, it might be under a high level of functional constraints because of its pivotal position in the pathway (first step of closure of the D-ring). A different isoform of *easA* in *C. africana* and *C. gigantean* reacts differently, creating a shunt yielding dihydroergot alkaloids (Figure 1). This diversification may result from the change in ecological niches. Nevertheless, the rates of diversity and divergence of *easA* were the second lowest after *easD*, even though it is physically located in between *lpsB* and *lpsC*. Both of these later step genes had much higher rates than *easA*, possibly due to fewer constraints or more direct positive selection, as they are involved in the final steps. The *cloA* gene represents another point of the pathway where shunts may take place. Presumably depending on the different isoforms of *cloA*, varied levels of oxidation occur, resulting in different end-products [13,15]. The high rates of diversity and divergence of *cloA* may reflect a high level of positive selection.

The signatures of selective pressure in DNA sequences could be detected through neutrality tests. For instance, if the value of Tajima’s D significantly deviates from zero, it indicates the presence of selective pressures, i.e., negative values suggest a positive selection, whereas positive values indicate balancing selection [58]. We conducted neutrality tests and found that none of the genes departed significantly from neutrality (results not shown). These results are contradictory to Liu et al. [59], in that *easE* and *easA* were under positive selection in Canadian and western USA *C. purpurea* populations. We speculate here that the small sample sizes in present study (28 sequences versus 200–300 in the previous study) might be the factor limiting the ability of the Tajima’s D test to detect selective pressures.

Compared with *eas* gene pathways, it is difficult to evaluate whether or not the evolutionary pattern of *ltm* genes conformed with the hourglass model because the sequential order of steps was uncertain. Even if we assume that paspaline-derived compounds are the main products, in the absence of *ltmG,* there are only two to three sequential steps to paspaline. Nevertheless, *ltmM* had the lowest rate of divergence and diversity compared with earlier (*ltmC*) and later steps (*ltmP* and *Q*).

Our results provide evidence for the first time that *eas* gene evolution follows the hourglass model. Whether this pattern exists in other metabolic gene pathways and the mechanisms that underpin this or other patterns are questions to be answered in future work.

## 4. Materials and Methods

### 4.1. Genome Aquisition

Fifty-four genomes of 19 *Claviceps* spp. were studied. The assemblies of 17 genomes and the raw reads of another 34 genomes were from previous studies (Table 1) [44,45], which outlined the protocols for the DNA extraction, library preparation, and sequencing platforms. In the present study, three additional genomes were sequenced (LM63, LM65, and LM72) using a protocol similar to that described in [44]. Briefly, the gDNA samples were normalized to 300 ng and sheared to 350 bp fragments using an M220 Covaris Focused-Ultrasonicator instrument (Covaris, Woburn, MA, USA). The obtained inserts were used as a template to construct PCR-free libraries using the NxSeq AmpFREE Low DNA Library kit (LGC, Biosearch Technologies, Middleton, WI, USA)) following LGC’s library protocol. Balanced libraries in equimolar ratios were pooled, and paired-end sequencing was carried on a NextSeq500/550 (Illumina, San Diego, CA, USA) using 2 × 150 bp NextSeq Mid Output Reagent Kit (Illumina, San Diego, CA, USA) according to the manufacturer’s recommendations.

The new assemblies of 37 genomes were achieved using the following protocols: raw reads were trimmed using BBDuk, a component of BBTools downloaded from the Joint Genome Institute website (https://jgi.doe.gov/data-and-tools/bbtools/ accessed on 9 November 2021). Both quality-trim and kmer-trim were applied using the parameters qtrim = rl, trimq = 20, forcetrimleft = 10, minlength = 36, ftm = 5, ref = adapters/adapters.fa, ktrim = r, k = 22, mink = 11, hdist = 1, tbo tpe. The qualities of initial reads and post-trimming reads were assessed using FastQC version 0.11.9, setting parameters as quiet, noextract. Pairs of trimmed reads for each strain were assembled using the SPAdes version 3.14.0 genome assembly toolkit with the default parameters [60]. QUAST version 5.0.2 was used to evaluate the resulting assemblies and to obtain statistics about the assembled contigs [61]. To assess the completeness of the genome assemblies, BUSCO 4.1.4 was run on the contigs using the fungal database (fungi odb10) (Creation date: 10 September 2020, number of species: 549, number of BUSCOs: 758) [62].

### 4.2. Alkaloid Gene Screening and Extraction

To investigate the presence/absence of the four classes of alkaloid synthesis genes in 54 genomes, BLAST searches were conducted to interrogate the genomes with the reference genes of interest using an in-house perl script (running blastn with an E-value of E^−99^ as the cutoff). Alternatively, each individual genome assembly was mapped onto the reference genes using the ‘Map to Reference’ function in Geneious prime 2020.1.2 (https://www.geneious.com, accessed on 9 November 2021). The reference gene clusters were downloaded from GenBank and applied as follows: the clusters of 14 ergot alkaloid synthesis (*eas*) genes and six indole-diterpene/lolitrem genes (*IDT/ltm*) from *C. purpurea* strain 20.1 (JN186799 containing *cloA*, *dmaW*, *easA*, *C*–*G*, *easH1*, *easH2*, *lpsA1*, *lpsA2*, *lpsB*, and *lpsC*; JX402756 containing *idt/ltmB*, *C*, *M*, *P*, *Q*, and *S*) and *C. paspali* RRC-1481 JN186800 (*easO*) were first applied as a query to interrogate each genome. In addition, the cluster from *C. fusiformis* PRL1980 EU006773 (10 genes: *cloA*, *dmaW*, *easA*, *C*–*H*, and *lpsB*) were applied to further interrogate genomes in *C.* sect. *Pusillae* and *C. citrina*. For the *IDT/ltm* genes that were not previously reported in *Claviceps purpurea* 20.1, the reference sequences from *C. paspali* JN613321 (*ltmF* and *ltmG*) and *Epichlo**ë* (*ltmE* and *J* on JN613318, and *K* on JN613320) were used to conduct lower stringency megablast searches (https://www.geneious.com, accessed in 9 November 2021) with E-values E^−50^ and E^−20^. Megablast searches were also conducted for loline alkaloid genes (*lolA*, *D*, *E*, *M–P*, *T*, and *U* on JF830816, *lolC* FJ464781, and *lolF* FJ594413) and peramine *(perA* JN640287) in all 54 genomes. Genes that were present in genomes were extracted manually. Split fragments of a single gene on different contigs were concatenated on the basis of reference sequences. DNA sequences of genes extracted from the new genomes were submitted to GenBank.

When multiple copies of certain genes were present (such as *dmaW*, *easE*, *easF*, *ltmB*, and *ltmQ*), the copy on the main cluster was designated as copy 1, as determined by examining the contig numbers. The exception was *easH,* which was determined on the basis of the similarity to the two copies determined by previous studies [14]. Disconnected fragments shorter than 300 bps were not considered.

### 4.3. Phylogenetic Analyses

The extracted sequences for each gene were aligned individually through the Geneious Prime (https://www.geneious.com, accessed on 9 November 2021) Align/Assemble function using Global alignment with free end gaps, 93% similarity (5.0/−9.026168) as the cost matrix, a gap open penalty of 12, a gap extension penalty of 3, and two refinement iterations. This protocol is particularly suitable for aligning sequences with large gaps or shorter fragments to full-length sequences. Maximum likelihood phylogenetic trees were developed using the PhyML 3.3.20180621 [63] plugin of Geneious Prime (https://www.geneious.com, accessed on 9 November 2021). Both GTR and HKY substitution models were attempted; branch supports were evaluated through bootstrapping analyses of 100 replicates. Reference sequences of *lpsB* of *C. paspali* has only 52% similarity with *C. purpurea*, causing spurious alignment and a significantly long branch; therefore, they were not included in the analyses.

### 4.4. Intraspecific Gene Diversity and Divergence Analyses

Population demographic parameters are suitable for investigating genetic differentiation and gene evolution at an intraspecific level. We investigated the DNA polymorphisms, nucleotide diversity (Pi), and average number of nucleotide differences (K) among 27 strains of *C. purpurea* using DnaSP [64]. Another reason for choosing this sub-set of data, instead of all 53 samples, is that all but three strains (LM65, LM2, and LM582 lacked *easH2*) contained all 12 genes, making the results more comparable. Nonetheless, the sequences with long gaps causing a significant reduction in alignment length in *dmaW* and *easF* were excluded from the DnaSP analyses. In addition, the tree-based diversity and divergence from the center of the tree (COT) were calculated through the web-based DIVEIN software (https://indra.mullins.microbiol.washington.edu/DIVEIN/diver.html, accessed on 9 November 2021) [65]. The following parameters were applied: GTR substitution model, optimized equilibrium frequencies, the best of NNI and SPR tree improvement, and topology + branch length tree optimization algorithm. For multicopy genes (*dmaW*, *easE*, *easF*, and *easH*), we calculated the parameters for each individual copy and combined them as one gene (Table 4).

## Figures and Tables

**Figure 1 toxins-13-00799-f001:**
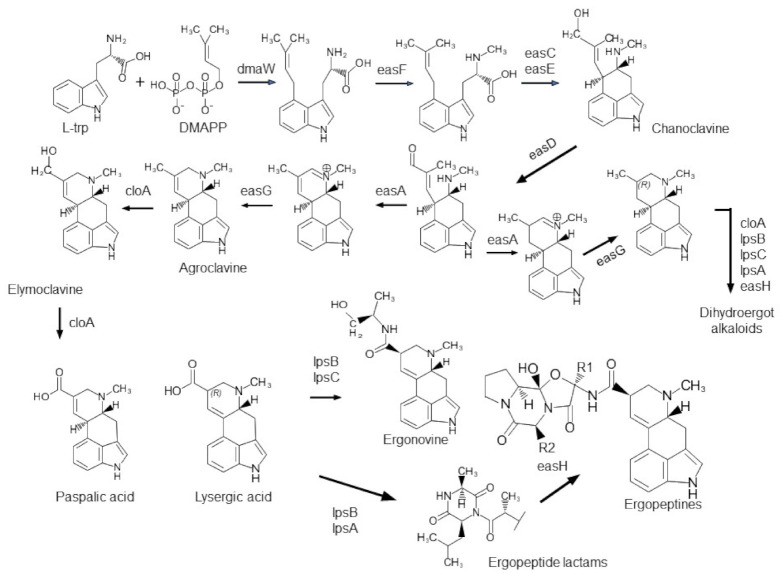
The ergot alkaloid biosynthetic pathway in *Claviceps* spp. Modified from Young et al. [13] and Robinson and Panaccione [15].

**Figure 2 toxins-13-00799-f002:**
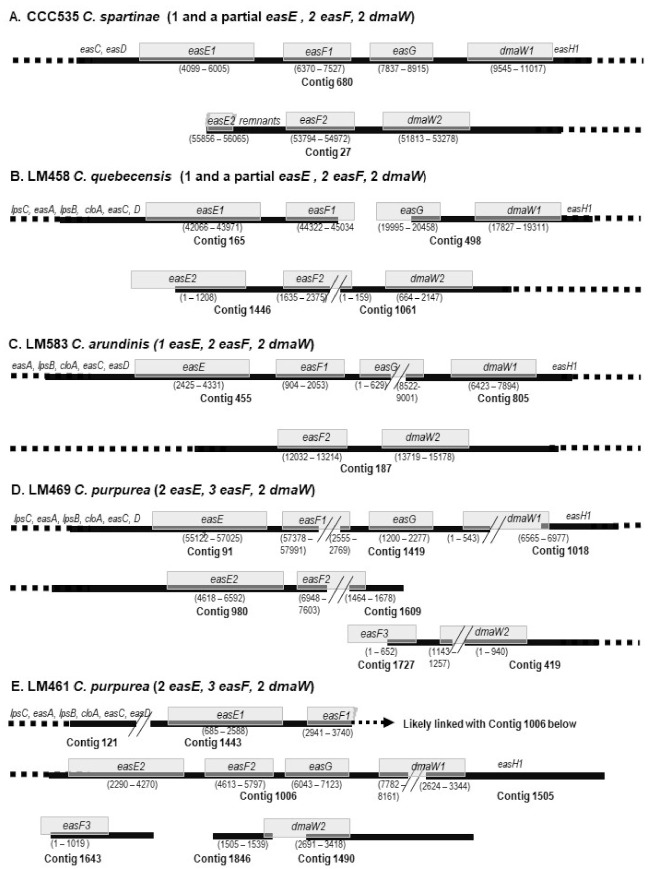
The schematic arrangements of multiple copies of *easE, easF*, and *dmaW* in relation to the primary cluster of other *eas* genes. The dark solid bars denote the contigs, while gray boxes represent genes labeled accordingly, with the ranges underneath. The lengths of genes and spaces are in approximate scale. The dashed bars and genes on them indicate that those genes are on the same contig; however, the details are not displayed. (**A**–**E**) represent different patterns of locations (see the text Section 2.2.1 for details).

**Figure 3 toxins-13-00799-f003:**
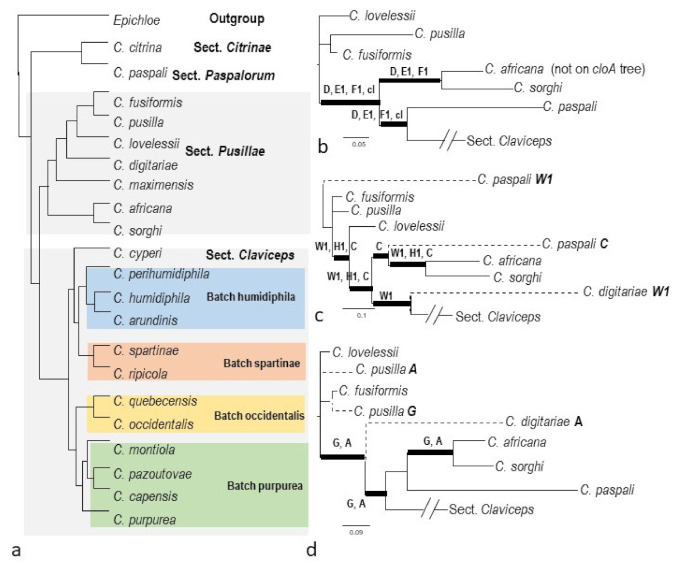
(**a**) The hypothetical species relationships of *Claviceps* spp. inferred by orthologous genes from Wyka et al. [44]. (**b**–**d**) Variant species relationships in Sect. *Pusillae* summarized from phylogenies inferred by each *eas* gene trees (Appendix A). The thickened branches denote bootstrapping values >80%. The letters next to thick branches denote the genes supporting the grouping, abbreviated as *A*, *C*—*H1* = *easA, easC*—*H1*; *cl* = *cloA*; *W1* = *dmaW1*. Dashed branches indicate that taxon was present on the gene trees listed after the species name. *lpsC* and *lpsB* are not listed here because only one or three sequences were available on the trees. DNA sequences of *C. fusiformis* and *C. paspali* were from GenBank EU006773 and JN613321.

**Figure 4 toxins-13-00799-f004:**
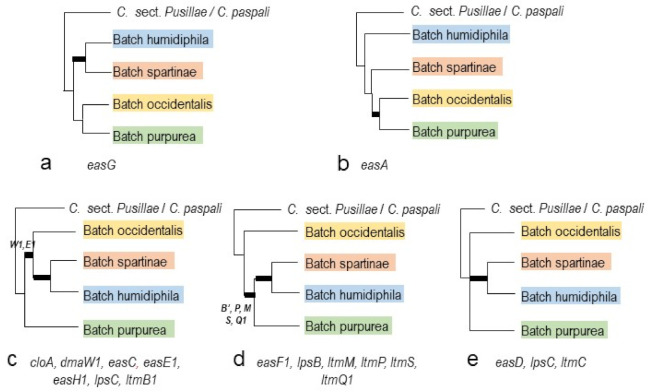
(**a**–**e**) Varied species relationships in sect. *Claviceps* summarized from phylogenetic trees of *eas* and *ltm* genes by PhyML analyses (the full trees are provided in Appendix A). The thick branches denote bootstrapping values >80%. The letters beside the thick branches indicate that those genes had strong support for those branches; otherwise, all genes listed below the figure had strong support.

**Figure 5 toxins-13-00799-f005:**
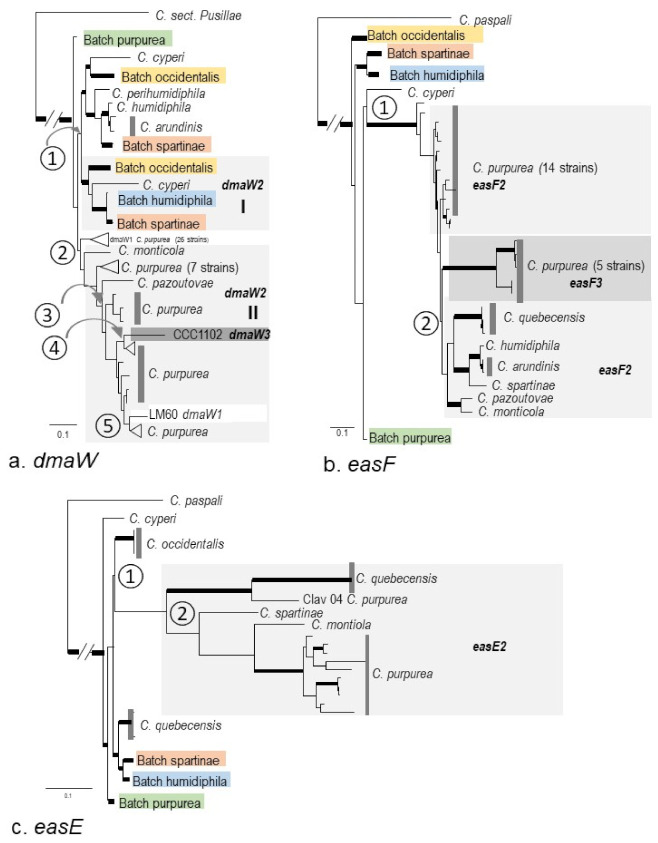
The simplified phylogenies of individual multicopy genes showing potential duplication events. The unedited trees generated by PhyML are presented in the Appendix A. (**a**) *dmaW*, (**b**) *easF*, (**c**) *easE*, (**d**) *easH*, (**e**) *ltmB*, and (**f**) *ltmQ*. The thickened branches indicate bootstrapping values ≥80%; dashed and hatches branches are shorter than their real length. The lineages that are not shaded gray are the first copies of each gene.

**Figure 6 toxins-13-00799-f006:**
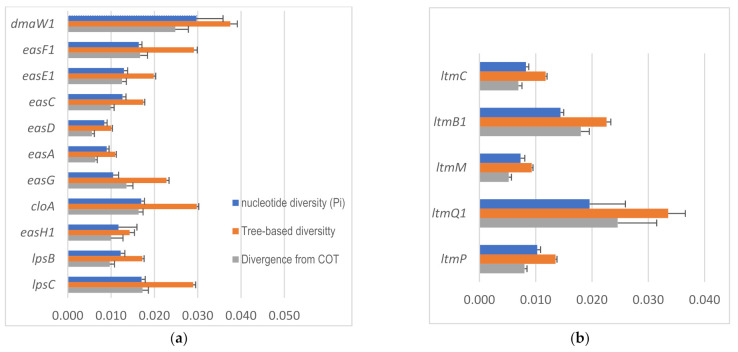
Nucleotide diversity and tree-based diversity and divergence for individual *eas* genes (**a**) and *idt/ltm* genes (**b**). Error bars denote the standard deviation for Pi and standard error for the other two parameters. The genes are arranged from top to bottom according to their order in the biosynthetic pathway. *ltmS* is not included in the chart as its function is unknown.

**Table 1 toxins-13-00799-t001:** Statistics of genome assemblies screened.

Species	Strain	BioSample	WGS #	Contigs	Total Length (bp)	Largest Contig (bp)	N50 (bp)	L50	GC (%)	Coverage (x)	Complete BUSCO’s (%)	
*C. arundinis*	CCC1102	SAMN11159893	JAIUSP000000000	1406	29,878,863	375,533	55,909	156	51.42	61x	98.1	
*C. capensis*	CCC1504	SAMN11159898	JAIUSL000000000	1497	27,462,555	202,637	39,758	198	51.69	66x	98.8	
*C. cyperi*	CCC1219	SAMN11159895	JAIUSO000000000	2467	26,149,012	130,032	19,946	386	51.72	56x	98.3	
*C. monticola*	CCC1483	SAMN11159896	JAIUSN000000000	1787	27,131,110	129,905	29,639	279	51.6	58x	98.8	
*C. occidentalis*	LM77	SAMN11159879	JAIURJ000000000	2285	28,557,246	118,746	21,556	410	51.37	58x	97.8	
*C. occidentalis*	LM84	SAMN11159876	JAIURI000000000	2119	28,639,296	133,797	23,641	389	51.39	164x	97.0	
*C. pazoutovae*	CCC1485	SAMN11159897	JAIUSM000000000	1619	27,544,752	151,196	35,477	229	51.7	61x	98.3	
*C. purpurea* s.s.	Clav04	SAMN11159846	JAIUSJ000000000	2581	30,594,081	349,533	29,378	296	51.69	46x	98.8	
*C. purpurea* s.s.	Clav26	SAMN11159847	JAIUSI000000000	1821	30,253,558	299,368	36,369	242	51.48	59x	98.8	
*C. purpurea* s.s.	Clav46	SAMN11159848	JAIUSG000000000	1887	30,292,940	231,314	36,582	246	51.08	58x	99.1	
*C. purpurea* s.s.	Clav52	SAMN11159849	JAIUSE000000000	1714	29,291,845	175,165	35,956	250	51.42	60x	98.9	
*C. purpurea* s.s.	Clav55	SAMN11159850	JAIUSD000000000	2023	30,195,775	203,523	33,461	261	51.55	59x	98.4	
*C. purpurea* s.s.	LM14	SAMN11159853	JAIUSC000000000	1888	30,259,282	163,532	32,812	268	51.74	49x	97.9	
*C. purpurea* s.s.	LM207	SAMN11159861	JAIUSB000000000	1910	30,165,540	260,847	31,428	273	51.74	53x	98.7	
*C. purpurea* s.s.	LM223	SAMN11159862	JAIURY000000000	1894	30,223,423	195,661	31,693	291	51.73	74x	98.4	
*C. purpurea* s.s.	LM232	SAMN11159863	JAIURX000000000	1911	30,304,653	216,996	33,376	265	51.73	53x	98.8	
*C. purpurea* s.s.	LM233	SAMN11159864	JAIURW000000000	1928	30,249,987	183,378	33,023	273	51.74	49x	98.3	
*C. purpurea* s.s.	LM30	SAMN11159855	JAIURV000000000	1816	30,203,936	160,353	35,005	265	51.75	64x	98.4	
*C. purpurea* s.s.	LM33	SAMN11159856	JAIURU000000000	2011	30,162,301	157,176	28,954	306	51.75	45x	97.9	
*C. purpurea* s.s.	LM39	SAMN11159857	JAIURT000000000	1797	30,183,718	168,047	34,902	258	51.75	81x	98.3	
*C. purpurea* s.s.	LM4	SAMN11159851	JAIURS000000000	1866	30,197,808	200,831	31,054	281	51.74	64x	98.3	
*C. purpurea* s.s.	LM46	SAMN11159858	JAIURR000000000	1842	30,109,785	205,399	32,503	270	51.76	79x	98.6	
*C. purpurea* s.s.	LM461	SAMN11159865	JAIURQ000000000	2041	30,157,824	190,247	28,928	307	51.74	37x	97.7	
*C. purpurea* s.s.	LM469	SAMN11159866	JAIURP000000000	1836	30,218,091	199,880	34,408	269	51.74	75x	98.3	
*C. purpurea* s.s.	LM470	SAMN11159867	JAIURO000000000	2482	30,086,038	123,014	23,231	384	51.75	26x	97.9	
*C. purpurea* s.s.	LM474	SAMN11159868	JAIURN000000000	1917	30,149,711	232,504	30,855	283	51.75	64x	98	
*C. purpurea* s.s.	LM5	SAMN11159852	JAIURM000000000	1817	30,171,863	188,144	34,174	271	51.74	67x	98.4	
*C. purpurea* s.s.	LM60	SAMN11159859	JAIURL000000000	1871	30,274,458	180,242	31,977	275	51.73	81x	98.6	
*C. purpurea* s.s.	LM63	SAMN20436330	JAIUSS000000000	1674	30,276,205	210,630	40,954	218	51.79	68x	98.4	
*C. purpurea* s.s.	LM65	SAMN20436331	JAIUSR000000000	1822	30,277,382	206,609	37,976	241	51.78	71x	98.4	
*C. purpurea* s.s.	LM71	SAMN11159860	JAIURK000000000	1919	30,241,564	172,997	32,324	282	51.76	168x	98	
*C. purpurea* s.s.	LM72	SAMN20436332	JAIUSQ000000000	1986	30,160,156	282,506	36,805	249	51.81	63x	98.4	
*C. quebecensis*	Clav32	SAMN11159882	JAIUSH000000000	1362	28,435,427	248,888	42,252	192	51.61	64x	99.0	
*C. quebecensis*	Clav50	SAMN11159881	JAIUSF000000000	1404	28,499,699	294,425	47,797	178	51.6	59x	98.8	
*C. ripicola*	LM219	SAMN11159874	JAIUSA000000000	1847	30,428,256	154,690	34,898	254	51.39	55x	97.1	
*C. ripicola* c.f.	LM220	SAMN11159873	JAIURZ000000000	1662	30,409,961	205,881	43,971	211	51.43	91x	97.7	
*C. spartinae*	CCC535	SAMN11159888	JAIUSK000000000	2017	28,974,645	142,723	28,332	300	51.38	60x	98.1	
Assemblies from previous studies												
*C. arundinis*	CCC1102	SAMN11159893	SRPS01	1406	29,878,863	375,533	55,909	156	51.42	61x	97.7 *	
*C. africana*	CCC489	SAMN11159887	SRPY01	5329	31,933,801	98,049	12,225	752	44.68	56x	95 *	
*C. arundinis*	LM583	SAMN08798359	QEQZ01	1613	30,055,381	164,904	39,306	223	51.42	69x	96.9	
*C. citrina*	CCC265	SAMN11159885	SRQA01	4830	25,056,896	81,802	8747	871	47.57	64x	92.2*	
*C. digitariae*	CCC659	SAMN11159892	SRPT01	3821	31,170,596	116,859	16,077	572	45.5	57x	95.9 *	
*C. humidiphila*	LM576	SAMN08798355	QERB01	1831	30,488,243	190,085	34,787	261	51.51	77x	97.9	
*C. lovelessii*	CCC647	SAMN11159891	SRPU01	8201	34,575,813	65,439	5747	1781	43.61	53x	91.6 *	
*C. maximensis*	CCC398	SAMN11159886	SRPZ01	2317	29,114,417	192,851	37,101	230	46.66	58x	98.3 *	
*C. occidentalis*	LM78	SAMN08800200	QEQY01	2321	28,571,683	125,459	21,416	422	51.37	64x	97.3	
*C. perihumidiphila*	LM81	SAMN08800226	QEQX01	1423	30,694,913	232,029	46,526	192	51.5	140x	96.9	
*C. purpurea* s.s.	LM28	SAMN08797627	QERD01	1930	30,251,797	260,842	31,815	274	51.74	49x	97.9	
*C. purpurea* s.s.	LM582	SAMN08798357	QERA01	2207	30,199,509	132,072	27,199	334	51.74	89x	98.6	
*C. pusilla*	CCC602	SAMN11159889	SRPW01	9171	37,319,484	83,555	5659	1917	41.84	52x	90.9 *	
*C. quebecensis*	LM458	SAMN08851611	QEQW01	1700	35,882,593	1,850,351	41,784	191	51.87	78x	98.0	
*C. ripicola*	LM454	SAMN08798353	QERC01	2108	30,692,668	189,162	28,587	314	51.37	156x	97.9	
*C. ripicola*	LM218	SAMN08798202	QERE01	1630	30,598,250	206,723	39,763	232	51.4	146x	97.6	
*C. sorghi*	CCC632	SAMN11159890	SRPV01	7206	31,897,900	112,296	6643	1389	45.24	60x	89.9 *	

* BUSCO completeness for these strains was based on the Dikaryon fungal database; see Wyka et al. [44] for details.

**Table 2 toxins-13-00799-t002:** The *eas* gene copies and their locations in 18 species in *Claviceps* sect. *Claviceps* and sect. *Pusillae*.

Section	Organism	Asbl *	Sample	*lpsC*		*easA*	*lpsB*	*cloA*	*easC*		*easD*		*easE*			*easF*					*easG*		*dmaW*					*easH*		
													E1		E2	F1		F2		F3			W1		W2		W3	H1		H2
*Claviceps*	*C. arundinis*	WF	CCC1102	418	/411	411	411	411	411		411		411			411		273			411		411	/583	273		707	583		
		SW	CCC1102			157	157	157	157		157		157			157		73			157		157		73		305	157		458
		BW	LM583			455	455	455	455		455		455			455		187			455	/805	805		187			805		822
	*C. capensis*	WF	CCC1504	173		173	173	173	173		173		173			173					173		1354					347		
	*C. cyperi*	WF	CCC1219	277		277	277	277	277		277		277			277					277		277	/2037	2094	/696		525		
	*C. humidiphila*	BW	LM576	599		259	259	259	259		259		259			259		390			259		259		390			259		701
	*C. monticola*	WF	CCC1483	367		367	367	367	367		367		367		986	367		986			367		367	/1745	986	/966		494		
	*C. occidentalis*	WF	LM77	202		202	202	202	202		202		202			202					202		1262		702			1887		1693
		BW	LM78	192		192	192	192	192		192		192			192					192		1273		722			1871		1675
		WF	LM84	290		290	290	290	290		290		290			290	/1715				1715		1340		721			1779		1618
	*C. pazoutovae*	WF	CCC1485	307		307	307	307	307		307		307			307		767			307		307	/1479	767	/622		430		
	*C. perihumidiphila*	BW	LM81			114	114	114	114		114		114			114					114	/604	604		359			604		710
	*C. purpruea*	WF	Clav52	131		131	131	131	131		131		131		739	131	/1395	739			1395		1395	/923	879			923		835
		WF	Clav04	71		71	71	71	71		71		71		1407	71		1514	/2293		71		71	/993	2459	/594		993		874
		WF	Clav26	105		105	105	105	105		105		105			105		816	/1596		105		105	/1453	1596			759		836
		WF	Clav46	201		201	201	201	201		201		201		1255	201	/1358	1255	/1668		1358		1358	/928	1481	/1294		928		1043
		WF	Clav55	419		419	419	419	419		419		419		1226	419	/1416	1226	/1797		1416		1416	/1316	1797	/1399		1316		1168
		WF	LM14	85		85	85	85	85		85		85			85					85		85	/699	1391	/539		699		716
		WF	LM207	374		374	374	374	374		374		374			374					374		374	/1169	1625	/1506		1169		1183
		WF	LM223	444		444	444	444	444		444		444			444	/1556	1027	/1718		1556		1556	/783	1460			783		762
		WF	LM232	112		112	112	112	112		112		112			112					112		112		1843			1148		908
		WF	LM233	89		89	89	89	89		89		89			89					89	/658	658		533			658		692
		BW	LM28	126		126	126	126	126		126		126			126					126		126		1874	/563		126		1208
		WF	LM30	88		88	88	88	88		88		88			88					88	/1022	1022		610			1022		1180
		WF	LM33	106		106	106	106	106		106		106			106					106		106	/1343	1781	/649		1343		1210
		WF	LM39	100		100	100	100	100		100		100		855	100	/1391	855	/1766	1340	1391		1391	/753	1340	/546		753		703
		WF	LM4	152		152	152	152	152		152		152			152					152		152	/1570	1797			857		726
		WF	LM46	209		209	209	209	209		209		209			209					209		209		1386			209		1025
		WF	LM461	121		121	121	121	121		121	/1443	1443		1006	1443		1006		1643	1006		1006	/1505	1846	/1490		1505		1402
		WF	LM469	91		91	91	91	91		91		91		980	91	/1419	980	/1609	1727	1419		1419	/1018	1727	/419		1018		680
		WF	LM470	294		294	294	294	294		294		294		2080	294	/1428	2080	/2165		1428		1428		2358			949		736
		WF	LM474	158		158	158	158	158		158		158		829	158	/1493	829	/1365		1493		1493	/788	1782	/41		788		719
		WF	LM5	121		121	121	121	121		121		121			121					121		121	/1217	1546	/679		1217		1198
		BW	LM582	98		98	98	98	98		98		98			98					98		98		1036			800		
		WF	LM60	333		333	333	333	333		333		333			333					333	/1791	1790	/1266	1480	/676		1266		1242
		WF	LM71	159		159	159	159	159		159		159			159					655		655		931			655		727
		WF	LM63	160		160	160	160	160		160		160		1178	160		1178		985	621	/160	621		985			621		918
		WF	LM65	21		21	21	21	21		21		21			21		1018		254	21		21		254			21		
		WF	LM72	190		190	190	190	190		190		190		1156	190		1156			5		5		912			5		
	*C. quebecensis*	WF	Clav32	308		308	308	308	308		308		308		753	308	/201	753	/730		201		201		730			201		
		WF	Clav50	227		227	227	227	227		227		227		731	227	/231	731	/679		231		231		679			231		
		BW	LM458	165		165	165	165	165		165		165		1446	165		1446	/1061		498		498		1061			498		
	*C. ripicola*	BW	LM218	81		81	81	81	81		81		81			81					81		81		527			81		820
		WF	LM219	78		78	78	78	78		78		78			78					78		78		533			78		
		WF	LM220	77		77	77	588	588		588		588			588					588		588		95			588		285
		BW	LM454			120	120	120	623		623		623			623					623		623		107			623		
	*C. spartinae*	WF	CCC535	1156	/1375	853	853	853	680		680		680		27	680		27			680		680		27			680		
*Pusillae*	*C. africana*	SW	CCC489			424			424		424		424			424					424		424					424	/1076	
	*C. digitariae*	SW	CCC659			403																	403							
	*C. lovelessii*	SW	CCC647			1632	1632	2885	2885	/4933	556	/4933	556			556					556		556					143		
	*C. maximensis*	WF	CCC398																											
	*C. pusilla*	SW	CCC602			3688	1435	3688	3968	/3891	3968		3968	/3180							3180		1918					1809		
	*C. sorghi*	SW	CCC632			2692		2475	2475		186		186			186					186		186					3370		

* The assembly versions: BW was from Wingfield et al. [45], SW was from Wyka et al. [44], and WF was generated in the present study; values in the cells denote contig numbers; the 2nd contig number was led by a/when the fragment is on two contigs; green color represents full-length genes, light orange represents partial or gapped sequences, and no fill represents no gene matches; hatches denote fragments containing frameshifts or internal stop codons. None of the genes were detected in *C. citrina* (*C.* sect. *Citrinae*, not listed).

**Table 3 toxins-13-00799-t003:** The *idt/ltm* gene copies and their locations in *C.* sect. *Claviceps* and sect. *Citrinae*.

Section	Organism	Assembly *		*ltmQ1*	*ltmQ2*	*ltmP*	*ltmB1*	*ltmB2*	*ltmC*	*ltmS*	*ltmM*	*ltmG*
*Citrinae*	*C. citrina*	WF	CCC265				1947		1947		582	2211
*Claviceps*	*C. arundinis*	WF	CCC1102	50		50	50	332	50	50	50	
		BW	LM583	158		158	158	124	158	158	158	
	*C. capensis*	WF	CCC1504	29		29	29		29	29	29	
	*C. cyperi*	WF	CCC1219			25	25		25		25	
	*C. humidiphila*	BW	LM576	945		945	478		745	745	745	
	*C. monticola*	WF	CCC1483	568		568	591		591	591	591	
	*C. occidentalis*	WF	LM77	1456	1898	1456	1538		1538	1538	657	
		BW	LM78	985	1877	985	985		985	985	691	
		WF	LM84	376	1789	376	376		376	376	376	
	*C. pazoutovae*	WF	CCC1485	225		225	185		185	185	185	
	*C. perihumidiphila*	BW	LM81	27		27	27	7	27	27	27	
	*C. purpruea*	WF	Clav52	174		174	174		174	174	1230	
		WF	Clav04	130	637	130	130		130	130	130	
		WF	Clav26	116		116	116		116	116	116	
		WF	Clav46	43	229	43	43		43	43	43	
		WF	Clav55	1358/1838/1286	1444	1286	557		557	557	557	
		WF	LM14	243		243	243		243	243	243	
		WF	LM207	255		255	255		255	255	255	
		WF	LM223	327		327	327		327	327	327	
		WF	LM232	315		315	315		315	315	315	
		WF	LM233	7		7	7		7	7	7	
		BW	LM28	258		258	258		258	258	258	
		WF	LM30	87		87	87		87	87	87	
		WF	LM33	51		51	51		51	51	51	
		WF	LM39	192		192	192		192	192	192	
		WF	LM4	361		361	361		361	1220	1220	
		WF	LM46	29		29	29		29	29	29	
		WF	LM461	529/65	1592	965	965		965	965	1500	
		WF	LM469	37		37	37		37	37	37	
		WF	LM470	646		787	787		787	787	787	
		WF	LM474	243		243	243		243	243	243	
		WF	LM5	17		17	17		17	17	17	
		BW	LM582	112		112	112		112	112	112	
		WF	LM60	765		765	765		765	765	440	
		WF	LM71	393		1283	977		977	977	977	
		WF	LM63	433		433	433		433	433	433	
		WF	LM65	406		406	406		406	406	406	
		WF	LM72	549/1151		1151	361		361	361	361	
	*C. quebecensis*	WF	Clav32	56		56	56		56	56	56	
		WF	Clav50	91		91	91		91	91	91	
		BW	LM458	536		536/1563	475		475	475	475	
	*C. ripicola*	BW	LM218	191		191	191	136	191	191	191	
		WF	LM219	395		395	638	589	638	638	638	
		WF	LM220	368		368	368	591	368	368	368	
		BW	LM454	138		138	764	949	764	764	764	
	*C. spartinae*	WF	CCC535	225		225	225	47	225	1212	1212	

* The assembly versions: BW was from Wingfield et al. [45], SW was from Wyka et al. [44], and WF was generated in the present study; values in the cells denote contig numbers, two values connected by/indicate the fragment was on two contigs; green color represents full-length genes, light orange represents partial or gapped sequences, and no fill represents no gene matches; hatches denote fragments containing frameshifts or internal stop codons. None of the *idt/ltm* genes were detected in *C.* sect. *Pusillae* except for two short fragments of *ltmG* from *C. maximensis* CCC398 and *C. digitariae* CCC659 by low stringency search, which are not listed (see also Section 2.2.2).

**Table 4 toxins-13-00799-t004:** Nucleotide polymorphism, tree-based divergence, and diversity of ergot alkaloid (*eas*) and indole-diterpene/lolitrm (*idt*/*ltm*) synthesis genes in *C. purpurea*.

Biosynthesis Genes	# of Sequences ^1^	Total # of Sites	# of Sites (Excluding Indel)	Segregating Sites	Ratio	# of Haplotypes	Haplotype (Gene) Diversity	Nucleotide Diversity		Average Number of Nucleotide Differences	Tree-Based Divergence from COT ^2^		Tree-Based Diversity		
	N	n°	n	s	s/n	h	Hd	Pi	Std dev	K	Mean	Std Error	Mean	Std. Error	
**Ergot Alkaloid (*eas*) Genes**															
*dmaW*	35	1516	921	196	0.213	32	0.995	0.055	0.004	50.523	0.062	0.004	0.086	0.002	
*dmaW1*	21	1480	938	154	0.164	19	0.99	0.030	0.006	27.886	0.025	0.003	0.038	0.002	
*dmaW2*	14	1516	923	102	0.111	13	0.989	0.042	0.004	39.11	0.042	0.008	0.065	0.004	
*easF*	32	1232	630	121	0.192	24	0.972	0.058	0.015	36.548	0.049	0.011	0.081	0.003	
*easF1*	25	1232	642	40	0.062	17	0.953	0.016	0.001	10.54	0.017	0.002	0.029	0.001	
*easF2*	8	1232	634	101	0.159	8	1	0.048	0.020	30.464	0.031	0.015	0.055	0.010	
*easE*	34	2283	1607	577	0.359	28	0.979	0.085	0.021	135.938	0.085	0.028	0.147	0.008	
*easE1*	28	1921	1891	112	0.059	22	0.968	0.013	0.001	24.526	0.013	0.001	0.020	0.000	
*easE2*	7	2283	1614	536	0.332	7	1	0.132	0.034	212.238	0.115	0.042	0.218	0.032	
*easC*	28	1508	1503	89	0.059	23	0.979	0.013	0.001	10.034	0.010	0.001	0.017	0.000	
*easD*	28	851	846	37	0.044	22	0.976	0.008	0.001	7.1510	0.006	0.001	0.010	0.000	
*easA*	28	1143	1143	51	0.045	21	0.966	0.009	0.001	10.286	0.006	0.000	0.011	0.000	
*easG*	28	1151	923	57	0.062	20	0.974	0.010	0.001	9.6720	0.014	0.001	0.023	0.001	
*cloA*	28	2754	2110	173	0.082	23	0.979	0.017	0.001	35.796	0.016	0.001	0.030	0.000	
*easH*	51	1195	569	247	0.434	23	0.936	0.132	0.014	75.151	0.102	0.013	0.154	0.004	
*easH1*	28	947	943	64	0.068	15	0.944	0.012	0.004	11.053	0.010	0.003	0.014	0.001	
*easH2*	23	1195	571	232	0.406	15	0.858	0.168	0.017	95.85	0.150	0.033	0.188	0.010	
*lpsB*	28	4006	3961	255	0.064	23	0.979	0.012	0.001	48.484	0.010	0.001	0.017	0.000	
*lpsC*	27	5431	5416	421	0.078	25	0.994	0.017	0.001	92.379	0.017	0.001	0.029	0.001	
**Indole-Diterpene/Lolitrem (*idt/ltm*) Genes**															
*ltmC*	28	1249	1246	60	0.048	25	0.992	0.008	0.000	10.299	0.007	0.001	0.012	0.000	
*ltmB1*	28	871	868	38	0.044	21	0.971	0.014	0.001	12.516	0.066	0.010	0.024	0.001	
*ltmM*	28	1766	1731	74	0.043	25	0.992	0.007	0.001	12.68	0.005	0.000	0.009	0.000	
*ltmQ*	25	2180	2119	293	0.138	24	0.997	0.019	0.006	41	0.026	0.010	0.040	0.004	
*ltmQ1*	24	2180	2119	292	0.138	23	0.996	0.020	0.006	41.486	0.025	0.007	0.034	0.003	
*ltmP*	28	1949	1895	111	0.059	24	0.981	0.010	0.001	19.479	0.008	0.000	0.014	0.000	
*ltmS*	28	955	924	34	0.037	24	0.989	0.007	0.001	6.839	0.008	0.001	0.015	0.000	

^1^ Sequences with large gaps causing a significant reduction in the number of sites were excluded from the analyses. ^2^ Tree-based divergence from the center of tree (COT) and diversity were estimated by DIVIEN; other parameters were estimated by DnaSP.

## Data Availability

The genome and gene data presented in this study are openly available in NCBI upon publication of this article https://www.ncbi.nlm.nih.gov/ (accessed on 9 November 2021). Accession numbers are detailed in the text Section 2.2 and Table 1.

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
