# Peer review of "Mining Indole Alkaloid Synthesis Gene Clusters from Genomes of 53 Claviceps Strains Revealed Redundant Gene Copies and an Approximate Evolutionary Hourglass Model"

_toxins, 2021, doi:10.3390/toxins13110799_

Round 1

Reviewer 1 Report

This manuscript investigated the genomes of 53 strains of 19 Claviceps spp. To understand the evolutionary patterns of the indole alkaloid synthesis genes by applying phylogenetic and DNA polymorphism analyses. The authors conducted very comprehensive analyses and also it claimed as the first study to examine the variations of each gene from the 28 strains of C. purpurea. I do not have specific arguments for this manuscript, but wonder how the authors convert the phylogenetic results to the proposed hourglass model? It seems not clear in the 3.3 section (page 23-24)

Author Response

Point one: How the authors convert the phylogenetic results to the proposed hourglass model? It seems not clear in the 3.3 section (page 23-24)

Answer: Thank you for the comments. The hourglass model was revealed by the divergence and diversity parameters resulted from analyses of sequence matrices using dnaSp and DEVIEN, which were presented in Table 4 and Figure 6. We agree with the reviewer that this was not clearly explained in the Discussion section (3.3). In the revised version, the sentences were slightly modified to provide a bit more clarification. I hope that helps.

Reviewer 2 Report

Title: Mining indole alkaloid synthesis gene clusters from genomes of 53 Claviceps strains revealed redundant gene copies and an approximate evolutionary hourglass model

Overview and general recommendation:

The article reported the presence of four classes of alkaloid genes (clusters) in 53 strains of 19 Claviceps species and the evolutionary patterns of these genes at inter- and intra-specific levels were also studied. In my opinion,  it is a well written manuscript that presented a suitable background and showed clear results as well as detailed methodologies.

  1. Тhe title and objectives are clear and interesting.
  2. The introduction is straightforward and provides enough information to get the reader up to speed.
  3. The procedures employed are well explained.
  4. Тhe figures and tables are well described. The quality of figs should be increased.
  5. The results and discussion are well described.
  6. Too much refs were cited.

Author Response

Point one: The quality of figs should be increased.

Answer: Thanks for pointing that out. The figures imbedded in the text are the small JPG version. The original figures are in Powerpoint. The resolution can be improved if that is what the reviewer meant. Communication with the editor is on going.

Point two: too much references were cited.

Answer: Thanks for the comments. In the revised version, we trimmed the citations as much as we can from 79 to 65. I apologize that I could not keep tracking of the trimming because of the technical difficulty in EndNote.